# CPGD: Toward Stable Rule-based Reinforcement Learning for Language Models

## Abstract

Recent advances in rule-based reinforcement learning (RL) have significantly improved the reasoning capability of language models (LMs) with rule-based rewards. However, existing RL methods—such as GRPO, REINFORCE++, and RLOO—often suffer from training instability, where large policy updates and improper clipping can lead to training collapse. To address this issue, we propose Clipped Policy Gradient Optimization with Policy Drift (CPGD), a novel algorithm designed to stabilize policy learning in LMs. CPGD introduces a policy drift constraint based on KL divergence to dynamically regularize policy updates, and leverages a clip mechanism on the logarithm of the importance-sampling ratio to prevent excessive policy updates. We provide theoretical justification for CPGD and demonstrate through empirical analysis that it mitigates the instability observed in prior approaches. Furthermore, we show that CPGD significantly improves performance while maintaining training stability. Our implementation balances theoretical rigor with practical usability, offering a robust alternative for RL in the post-training of LMs.

## 1 Introduction

Rule-based reinforcement learning (RL) has emerged as a key approach for eliciting reasoning capabilities in language models (LMs) (DeepSeek-AI et al., 2025). It leverages simple, efficient reward functions derived from deterministic rules, effectively mitigating reward hacking (Gao et al., 2022) while activating reasoning abilities of models (DeepSeek-AI et al., 2025; Polu & Sutskever, 2020; Le et al., 2022; Shinn et al., 2023). This has sparked a line of research focused on developing more effective RL algorithms for both textual and general multimodal reasoning tasks. Notable methods include GRPO (DeepSeek-AI et al., 2025), REINFORCE++ (Hu et al., 2025a), RLOO (Kool et al., 2019; Ahmadian et al., 2024), and GRPO variants such as DAPO (Yu et al., 2025), Dr.GRPO (Liu et al., 2025), and GPG (Chu et al., 2025). However, we observe that these RL methods often suffer from training instability, which we attribute to the use of the *importance-sampling ratios* in their loss functions. Although PPO-clip loss (Schulman et al., 2017) is commonly adopted to mitigate extreme policy updates, its one-sided nature fails to constrain large ratios when the advantage is negative—potentially causing gradient explosions dominated by poor samples, leading to catastrophic training collapse. We theoretically show that incorporating the ratio in the loss can amplify the policy shift, and our empirical results confirm that this can lead to training collapse in existing RL methods.

To address this issue, we propose *Clipped Policy Gradient Optimization with Policy Drift* (CPGD), an algorithm that replaces the PPO-clip loss with the REINFORCE loss (Sutton & Barto, 1998) to avoid instability caused by directly involving policy ratios in the loss function. To ensure proximal optimization, we introduce PPO's clip mechanism and a policy drift regularizer, constraining optimization within a local region and mitigating over-optimization that may impair reasoning behaviors as shown in Section 4.2. Furthermore, we develop a novel KL estimator that ensures correct corrective gradient directions while avoiding the potential numerical instability associated with the commonly used $k_3$ estimator (Schulman, 2023). We also incorporate weighted advantages to dynamically adjust the influence of each sample, further enhancing model performance. Furthermore, we theoretically prove the *monotonic improvement property* of CPGD and empirically demonstrate its superior training stability and performance.

Our experimental results show that CPGD consistently outperform the popular RL algorithms and strong open-source baselines across standard reasoning benchmarks. Notably, CPGD enhances overall performance by 10% across all benchmarks compared to the QwenVL2.5-7B/32B model in multimodal settings, whereas GRPO achieves a 4% improvement. Specially, compared to QwenVL2.5-7B, CPGD achieves +24.6% gain on the in-domain benchmark MMK12, and improves by +8.2% and +6.3% on the out-of-distribution benchmarks MathVista and MathVision, respectively. A similar trend in performance gains is also observed for InternVL2.5-8B (multimodal) and Qwen3-8B (text-only), demonstrating the superior generalization and enhancement capabilities of CPGD.

## 2 RELATED WORK

**RL for training reasoning models.** RL has become a key method for improving reasoning in LMs (DeepSeek-AI et al., 2025; OpenAI, 2024). While early methods rely on PPO (Schulman et al., 2017), its high computational cost has driven interest in alternatives like DPO (Rafailov et al., 2023), which simplifies training but depends on high-quality offline data. Recent RL methods such as GRPO, RLOO, and REINFORCE++ aim to balance stability and efficiency. Notably, DeepSeek R1 (DeepSeek-AI et al., 2025) shows that pure RL can elicit self-reflection and reasoning in LMs without SFT. Concurrent works have introduced GRPO variants to address its shortcomings. Dr.GRPO (Liu et al., 2025) identifies optimization bias in GRPO that favors longer response among incorrect ones. DAPO (Yu et al., 2025) offers improvements including decoupled clipping thresholds and token-level losses. GPG (Chu et al., 2025), in contrast, adopts a minimalist design by discarding both clipping and KL regularization, relying solely on the REINFORCE loss (Sutton & Barto, 1998). However, none of these approaches focus on the training instability issue in existing RL methods, which is the primary focus of this work. Concurrent research by leading teams such as MiniMax has also identified this instability phenomenon and proposed similar algorithms to address it, emphasizing the significance of the issue (MiniMax et al., 2025). We refer readers to the research (Zhang et al., 2025) for a more comprehensive survey.

**Large reasoning model.** Recently, a surge of reasoning models has emerged, driven by the principle of test-time scaling laws, which demonstrate that models with explicit reasoning processes achieve superior performance (Chen et al., 2025b). Leading models in this area include DeepSeek R1 (DeepSeek-AI et al., 2025), OpenAI's o-series (OpenAI, 2024), Qwen series (Team, 2025; 2024), and Kimi k1.5 (Team et al., 2025). However, their training pipelines and datasets remain undisclosed. This has motivated a wave of academic research within the open-source community, including parallel efforts such as OpenR1 (Face, 2025), TinyZero (Pan et al., 2025), LMM-R1 (Peng et al., 2025), R1-V (Chen et al., 2025a), Reason-RFT (Tan et al., 2025), and MM-Eureka (Meng et al., 2025). These works primarily focus on constructing high-quality datasets and complete training pipelines. They commonly adopt GRPO to enhance reasoning capabilities but do not specifically investigate improvements to the RL algorithms themselves.

## 3 PRELIMINARIES

### 3.1 PROBLEM FORMULATION

We denote an LM by $\pi_\theta$, where $\theta \in \mathbb{R}^d$ represents the model parameters. Given a prompt $\mathbf{x} = [x_1, \ldots, x_m] \in \mathcal{D}$, the model generates a response $\mathbf{y} = [y_1, \ldots, y_n]$ by sampling from the conditional distribution $\pi_\theta(\cdot|\mathbf{x})$, with both $x_i$ and $y_i$ drawn from a finite vocabulary $\mathcal{V}$. In this work, we focus on transformer-based LMs that generate responses autoregressively, such that $\pi_\theta(\mathbf{y}|\mathbf{x}) = \prod_{i=1}^{n} \pi_\theta(y_i|\mathbf{x}, \mathbf{y}_{<i})$, where $\mathbf{y}_{<i} = [y_1, \ldots, y_{i-1}]$ and $\mathbf{y}_{<1}$ is an empty sequence.

RL in post-training is typically modeled as a Markov decision process (MDP), defined by a tuple $\mathcal{M} = (\mathcal{S}, \mathcal{A}, \mathcal{P}, \mathcal{R}, \rho)$, where $\mathcal{S}$ is the state space, $\mathcal{A}$ is the action space, $\mathcal{P}$ is the transition kernel, $\mathcal{R}$ is the deterministic reward function, and $\rho$ is the initial state distribution. For LMs, two MDP formulations are widely considered: *token-level MDP* and *response-level MDP*. In a *token-level MDP*, each token is treated as a single action. At the time step $t$, the state $\mathbf{s}_t = [\mathbf{x}, \mathbf{y}_{<t}]$ includes the prompt and the tokens generated so far. The action $a_t = y_t$ is sampled according to $y_t \sim \pi_\theta(\cdot|\mathbf{x}, \mathbf{y}_{<t})$, where the action space $\mathcal{A}$ is equal to the vocabulary $\mathcal{V}$. The environment transitions deterministically to $\mathbf{s}_{t+1} = [\mathbf{x}, \mathbf{y}_{<t+1}]$. The reward is defined as $\mathcal{R}(\mathbf{s}_t, a_t) = \mathcal{R}([\mathbf{x}, \mathbf{y}_{<t}], y_t)$, and

$\rho$ is induced by the prompt distribution in $\mathcal{D}$. In a *response-level MDP*, the full response is treated as an individual action: $\mathbf{a} = \mathbf{y} \sim \pi_\theta(\cdot|\mathbf{x})$. The state is defined solely by the prompt $\mathbf{s} = \mathbf{x}$, and the episode terminates after one step. Thus, the transition kernel is omitted in the single-turn dialogue setting. The reward is $\mathcal{R}(\mathbf{s}, \mathbf{a}) = \mathcal{R}(\mathbf{x}, \mathbf{y})$, with $\rho$ again determined by $\mathcal{D}$.

### 3.2 PPO LOSS VS. REINFORCE LOSS

In RL, PPO loss and REINFROCE loss are two widely used policy optimization objectives. The REINFORCE loss is a direct and theoretically grounded approach derived from the policy gradient theorem (Sutton & Barto, 1998), which is expressed as:

$$\mathcal{L}(\theta) := \mathbb{E}_{\mathbf{x}\sim\mathcal{D},\mathbf{y}\sim\pi_{\theta_{old}}(\cdot|\mathbf{x})} \left[ \frac{1}{|\mathbf{y}|} \sum_{i=1}^{|\mathbf{y}|} A_i \cdot \ln \pi_\theta(y_i|\mathbf{x}, \mathbf{y}_{<i}) \right],$$

where $A_i$ is the advantage estimate for the $i$-th token. While simple and theoretically sound, REINFORCE loss suffers from high variance and unstable learning due to unbounded policy updates.

To mitigate such instability, Schulman et al. (2017) introduces two proximity-constrained variants: PPO-KL loss and PPO-clip loss. The former adds a KL divergence between the old and new policies:

$$\mathcal{L}(\theta) := \mathbb{E}_{\mathbf{x}\sim\mathcal{D},\mathbf{y}\sim\pi_{\theta_{old}}(\cdot|\mathbf{x})} \left[ \frac{1}{|\mathbf{y}|} \sum_{i=1}^{|\mathbf{y}|} \frac{\pi_\theta(y_i|\mathbf{x}, \mathbf{y}_{<i})}{\pi_{\theta_{old}}(y_i|\mathbf{x}, \mathbf{y}_{<i})} A_i - \alpha \cdot \hat{D}_{\text{KL}}(\theta_{old}, \theta) \right],$$

where the KL estimate $\hat{D}_{\text{KL}}(\theta_{old}, \theta) := \ln \frac{\pi_{\theta_{old}}(y_i|\mathbf{x},\mathbf{y}_{<i})}{\pi_\theta(y_i|\mathbf{x},\mathbf{y}_{<i})}$ corresponds to the $k_1$ estimator (Schulman, 2023). In addition to the $k_1$ estimator, the $k_3$ estimator, which takes the form $k_3(\theta_{old}, \theta) := \frac{\pi_\theta(y_i|\mathbf{x},\mathbf{y}_{<i})}{\pi_{\theta_{old}}(y_i|\mathbf{x},\mathbf{y}_{<i})} - 1 - \ln \frac{\pi_\theta(y_i|\mathbf{x},\mathbf{y}_{<i})}{\pi_{\theta_{old}}(y_i|\mathbf{x},\mathbf{y}_{<i})}$, is unbiased and exhibits lower variance.

On the other hand, the PPO-clip loss introduces a clipped surrogate objective that implicitly limits the magnitude of policy updates without requiring a KL term. The objective is defined as:

$$\mathcal{L}(\theta) := \mathbb{E}_{\mathbf{x}\sim\mathcal{D},\mathbf{y}\sim\pi_{\theta_{old}}(\cdot|\mathbf{x})} \left[ \frac{1}{|\mathbf{y}|} \sum_{i=1}^{|\mathbf{y}|} \min \left( \frac{\pi_\theta(y_i|\mathbf{x}, \mathbf{y}_{<i})}{\pi_{\theta_{old}}(y_i|\mathbf{x}, \mathbf{y}_{<i})} A_i, \text{clip}_{1-\epsilon}^{1+\epsilon} \left( \frac{\pi_\theta(y_i|\mathbf{x}, \mathbf{y}_{<i})}{\pi_{\theta_{old}}(y_i|\mathbf{x}, \mathbf{y}_{<i})} \right) A_i \right) \right],$$

$$(1)$$

where $\epsilon \in [0, 1]$, and $\text{clip}_a^b(x) := \max(\min(x, b), a)$.

### 3.3 RULE-BASED REINFORCEMENT LEARNING

This work focuses on verifiable tasks, where the outcome reward is determined by the final accuracy. Specifically, a response $\mathbf{y}$ receives a reward of 1 if it is the correct answer to the prompt $\mathbf{x}$, and 0 otherwise. We denote this reward function as $\mathcal{R}_o$ to emphasize its nature as an outcome-based reward. Within this setting, REINFORCE-style algorithms are favored as they reduce computational cost by forgoing critic networks. Notable methods include REINFORCE++ (Hu et al., 2025a), RLOO (Kool et al., 2019; Ahmadian et al., 2024), and GRPO (DeepSeek-AI et al., 2025).

**REINFORCE++:** REINFORCE++ enhances the standard REINFORCE framework by integrating key optimizations from PPO, improving both stability and efficiency. REINFORCE++ replaces the objective from REINFORCE loss with PPO-clip loss (Equation 1), and the advantage value is:

$$A_i^{\text{R++}} := \text{GlobalNorm}\left( G(\mathbf{x}, \mathbf{y}_{\le i}) \right), \quad G(\mathbf{x}, \mathbf{y}_{\le i}) := \mathcal{R}_o(\mathbf{x}, \mathbf{y}) - \beta \sum_{j=i}^{|\mathbf{y}|} \ln \frac{\pi_{\theta_{old}}(y_j|\mathbf{x}, \mathbf{y}_{<j})}{\pi_{\text{ref}}(y_j|\mathbf{x}, \mathbf{y}_{<j})}.$$

Here, $\ln \frac{\pi_{\theta_{old}}}{\pi_{\text{ref}}}$ is the KL penalty used to restrict the current policy from deviating too far from the reference policy $\pi_{\text{ref}}$ (typically the initial model $\pi_0$) to maintain stability. $\text{GlobalNorm}(x) := \frac{x - \text{mean}(\{x' \in \text{batch}\})}{\text{std}(\{x' \in \text{batch}\})}$ is the normalization operation across the global batch for all prompts.

**RLOO:** The primary distinction between RLOO and REINFORCE++ lies in their computation of the advantage value. RLOO first generates a group of $K$ responses $\{\mathbf{y}^{(k)}\}_{k=1}^K$ for each prompt $\mathbf{x}$

and computes the advantage using a *leave-one-out* strategy to reduce the gradient variance:

$$A_{i,k}^{\text{RLOO}} := \text{GlobalNorm}\left(\tilde{G}(\mathbf{x}, \mathbf{y}_{\leq i}^{(k)})\right), \text{ where } \tilde{G}(\mathbf{x}, \mathbf{y}_{\leq i}^{(k)}) := G(\mathbf{x}, \mathbf{y}_{\leq i}^{(k)}) - \frac{1}{K-1} \sum_{k' \neq k} G(\mathbf{x}, \mathbf{y}_{\leq i}^{(k')}).$$

**GRPO:** GRPO introduces a group-based advantage and employs an external KL divergence between the new policy $\pi_\theta$ and a reference policy $\pi_{\text{ref}}$ via the $k_3$ estimator. The loss is:

$$\mathcal{L}_{\text{GRPO}}(\theta; \theta_{old}) = \mathbb{E}_{\mathbf{x} \sim \mathcal{D}, \{\mathbf{y}^{(k)}\}_{k=1}^K \sim \pi_{\theta_{old}}(\cdot|\mathbf{x})} \left[ \frac{1}{K} \sum_{k=1}^K \left( \frac{1}{|\mathbf{y}^{(k)}|} \sum_{i=1}^{|\mathbf{y}^{(k)}|} \left( -\beta \cdot \mathcal{M}_{\theta,\text{ref}}^i(\mathbf{x}, \mathbf{y}^{(k)}) \right. \right. \right.$$
$$\left. \left. \left. + \min\left( \frac{\pi_\theta(y_i^{(k)}|\mathbf{x}, \mathbf{y}_{<i}^{(k)})}{\pi_{\theta_{old}}(y_i^{(k)}|\mathbf{x}, \mathbf{y}_{<i}^{(k)})} A_k^{\text{GRPO}}, \text{clip}_{1-\epsilon}^{1+\epsilon}\left( \frac{\pi_\theta(y_i^{(k)}|\mathbf{x}, \mathbf{y}_{<i}^{(k)})}{\pi_{\theta_{old}}(y_i^{(k)}|\mathbf{x}, \mathbf{y}_{<i}^{(k)})} \right) A_k^{\text{GRPO}} \right) \right) \right],$$

where

$$A_k^{\text{GRPO}} := \text{GroupNorm}(\mathcal{R}_o(\mathbf{x}, \mathbf{y}^{(k)})) = \frac{\mathcal{R}_o(\mathbf{x}, \mathbf{y}^{(k)}) - \text{mean}(\{\mathcal{R}_o(\mathbf{x}, \mathbf{y}^{(k)})\}_{k=1}^K)}{\text{std}(\{\mathcal{R}_o(\mathbf{x}, \mathbf{y}^{(k)})\}_{k=1}^K)},$$

$$\mathcal{M}_{\theta,\text{ref}}^i(\mathbf{x}, \mathbf{y}^{(k)}) := \frac{\pi_{\text{ref}}(y_i^{(k)}|\mathbf{x}, \mathbf{y}_{<i}^{(k)})}{\pi_\theta(y_i^{(k)}|\mathbf{x}, \mathbf{y}_{<i}^{(k)})} - 1 - \ln \frac{\pi_{\text{ref}}(y_i^{(k)}|\mathbf{x}, \mathbf{y}_{<i}^{(k)})}{\pi_\theta(y_i^{(k)}|\mathbf{x}, \mathbf{y}_{<i}^{(k)})}.$$

# 4 THE PROPOSED METHOD

This section introduces our RL algorithm, *Clipped Policy Gradient Optimization with Policy Drift* (CPGD), designed to improve the stability of RL training. In Section 4.1, we present the CPGD algorithm along with its theoretical guarantees, and highlight potential limitations of the standard PPO-clip loss. In Section 4.2, we provide empirical evidence of instability in existing methods and analyze its possible causes, showing how CPGD addresses them for more stable training. Finally, Section 4.3 describes the practical implementation of CPGD, striking a balance between theoretical soundness and practical implementation.

## 4.1 CLIPPED POLICY GRADIENT OPTIMIZATION WITH POLICY DRIFT (CPGD)

Under the response-level MDP assumption, CPGD aims to maximize the following formula:

$$\mathcal{L}_{\text{CPGD}}(\theta; \theta_{old}) = \mathbb{E}_{\mathbf{x} \sim \mathcal{D}} \left[ \mathbb{E}_{\mathbf{y} \sim \pi_{\theta_{old}}(\cdot|\mathbf{x})} \left[ \Phi_\theta(\mathbf{x}, \mathbf{y}) \right] - \alpha \cdot D_{\text{KL}}( \pi_{\theta_{old}}, \pi_\theta | \mathbf{x}) \right], \quad (2)$$

where

$$\Phi_\theta(\mathbf{x}, \mathbf{y}) := \min\left( \ln \frac{\pi_\theta(\mathbf{y}|\mathbf{x})}{\pi_{\theta_{old}}(\mathbf{y}|\mathbf{x})} \cdot A^{\text{CPGD}}(\mathbf{x}, \mathbf{y}), \text{clip}_{\ln(1-\epsilon)}^{\ln(1+\epsilon)}\left( \ln \frac{\pi_\theta(\mathbf{y}|\mathbf{x})}{\pi_{\theta_{old}}(\mathbf{y}|\mathbf{x})} \right) A^{\text{CPGD}}(\mathbf{x}, \mathbf{y}) \right),$$

$$A^{\text{CPGD}}(\mathbf{x}, \mathbf{y}) := \mathcal{R}_o(\mathbf{x}, \mathbf{y}) - \mathbb{E}_{\mathbf{y}' \sim \pi_\theta(\cdot|\mathbf{x})}\left[ \mathcal{R}_o(\mathbf{x}, \mathbf{y}') \right],$$

$$D_{\text{KL}}(\pi_{\tilde{\theta}}, \pi_\theta | \mathbf{x}) := \mathbb{E}_{\mathbf{y} \sim \pi_{\tilde{\theta}}(\cdot|\mathbf{x})}\left[ \ln \frac{\pi_{\tilde{\theta}}(\mathbf{y}|\mathbf{x})}{\pi_\theta(\mathbf{y}|\mathbf{x})} \right].$$

Hereinafter, we term the KL divergence between the old and current policies as *policy drift*, and between the current and reference policies as *reference constraint*. CPGD differs from the standard PPO-clip loss in two key aspects: (1) REINFORCE loss ($\ln \frac{\pi_\theta(\mathbf{y}|\mathbf{x})}{\pi_{\theta_{old}}(\mathbf{y}|\mathbf{x})}$) with the PPO-clip's clip mechanism is used. (2) A PPO-KL like policy drift is introduced, imposing a forward KL divergence penalty between the old and current policies $D_{\text{KL}}( \pi_{\theta_{old}}, \pi_\theta | \mathbf{x})$.

**Why use the REINFORCE loss?** In the original PPO objective, although the importance-sampling ratio corrects for the distribution mismatch between the old and current policies, it simultaneously introduces high variance. As empirically demonstrated in Section 4.2, such variance can destabilize training and even cause training collapse, while using a REINFORCE loss without the ratio substantially improves training stability. In Proposition 1 below, we further provide a theoretical

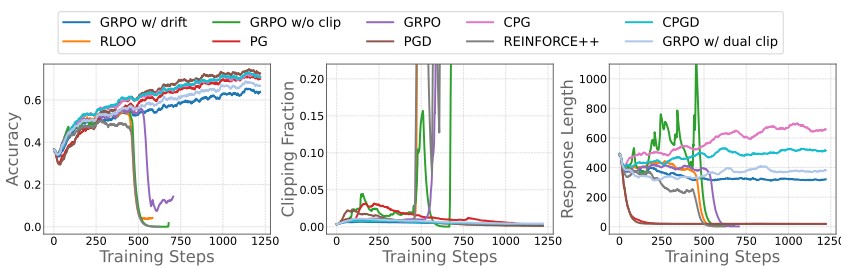

Figure 1: Accuracy, clipping fraction and response length curves throughout training.

explanation for this phenomenon, showing that the use of the policy ratio amplifies policy drift, causing the updated policy to exceed the intended bounds.

**Why introduce the policy drift and clip mechanism?** The clip mechanism and policy drift are introduced to enforce proximal policy updates, which are critical for the monotonic improvement guarantees in Theorem 1, and for mitigating reward hacking behaviors such as length collapse (see Section 4.2). Crucially, the clip mechanism reduces the need for a large weight on the policy drift term. When the policy stays within the clipping range, the drift term remains small, allowing the algorithm to focus on optimizing the main objective $\Phi$. If the policy strays beyond the range, the main objective's gradient is clipped to zero, and the drift term takes over to correct the deviation.

**Proposition 1.** *Let $\theta_0$ be a parameter such that the importance-sampling ratio satisfies $\left|\frac{\pi_{\theta_0}(\mathbf{y}|\mathbf{x})}{\pi_{\theta_{old}}(\mathbf{y}|\mathbf{x})} - 1\right| = \epsilon$. Consider updating $\theta_0$ using either (i) the PPO-clip objective, resulting in parameter $\theta_1^{PPO}$, or (ii) the CPGD objective with $\alpha = 0$ (denoted as CPG), yielding parameter $\theta_1^{CPG}$. Then, there exists a constant $\eta_{\max} > 0$ such that for any learning rate $\eta \in (0, \eta_{\max})$, the following inequality holds:*

$$\left|\frac{\pi_{\theta_1^{PPO}}(\mathbf{y}|\mathbf{x})}{\pi_{\theta_{old}}(\mathbf{y}|\mathbf{x})} - 1\right| > \left|\frac{\pi_{\theta_1^{CPG}}(\mathbf{y}|\mathbf{x})}{\pi_{\theta_{old}}(\mathbf{y}|\mathbf{x})} - 1\right| > \left|\frac{\pi_{\theta_0}(\mathbf{y}|\mathbf{x})}{\pi_{\theta_{old}}(\mathbf{y}|\mathbf{x})} - 1\right| = \epsilon.$$

*After one update step, both PPO and CPG increase the importance-sampling ratio deviation from the old policy, but PPO does so more aggressively than CPG.*

The following theorem further presents that CPGD enjoys the monotonic improvement guarantee, indicating its theoretical rationality. See Appendix B for the proofs of Proposition 1 and Theorem 1.

**Theorem 1.** *Let $\{\pi_{\theta_k}\}_{k=0}^{\infty}$ denote the sequence of policies generated by the CPGD update rule: $\theta_{k+1} = \arg\max_\theta \mathcal{L}_{CPGD}(\theta; \theta_{old})$ where the advantage function is competed as $A^{CPGD}(\mathbf{x}, \mathbf{y}) = \mathcal{R}_o(\mathbf{x}, \mathbf{y})$. Then, $\pi_{\theta_{k+1}}$ is better than $\pi_{\theta_k}$, i.e., $\eta(\theta_{k+1}) \geq \eta(\theta_k)$, where $\eta(\theta) := \mathbb{E}_{\mathbf{x} \sim \mathcal{D}, \mathbf{y} \sim \pi_{\theta_{old}}(\cdot|\mathbf{x})}[\mathcal{R}_o(\mathbf{x}, \mathbf{y})]$.*

### 4.2 TRAINING COLLAPSE

Several studies suggest that the reference constraint may hinder policy improvement (Yu et al., 2025; Hu et al., 2025b). However, we observe that removing this KL term leaves the PPO-clip loss alone insufficient to effectively constrain large policy shifts, which can lead to training collapse. While such collapse may be partially mitigated through techniques such as early stopping or small learning rates, it remains a latent instability that undermines the reliability of continued training. In this subsection, we examine training collapse and show that CPGD effectively prevents it.

Figure 1 presents training curves on the MMK12 dataset (Meng et al., 2025) for RLOO, REIN-FORCE++, GRPO, GRPO w/o clip (i.e., GRPO without the clip mechanism), GRPO w/ dual clip (i.e., the policy ratio is additionally clipped to no more than a constant—3.0 in our case—when advantage is negative (Ye et al., 2020)), GRPO w/ drift (i.e., GRPO with policy drift), PG (basic policy gradient), CPG (PG with the clip mechanism), PGD (PG with the policy drift), and CPGD, all without the reference constraint. We use QwenVL2.5-7B (Bai et al., 2023) as the base model. All algorithms share the same hyperparameters: a training and rollout batch size of 128, 8 responses per prompt, a learning rate of $1e{-}6$, one PPO epoch, and ten training episodes.

As shown in Figure 1, methods such as REINFORCE++, RLOO, GRPO w/o clip, and GRPO exhibit highly unstable policy ratio dynamics, leading to training collapse in mid stages. In contrast, GRPO

w/ dual clip, GRPO w/ drift, PG, CPG, PGD, and CPGD maintain stable training curves. GRPO w/ dual clip mitigates instability by globally constraining the policy ratio, while the PG series sidesteps ratio-induced variance by excluding it from the loss computation. These comparisons indicate that incorporating policy ratios in the loss can introduce high variance during fluctuations, and that simple one-sided clipping fails to recover from extreme ratios, ultimately causing collapse. Although dual clip mechanism stabilizes training, it may introduce new issues: frequent zero-gradient updates and ineffective learning under negative advantages due to the zero-gradient clipped large ratios. Additionally, GRPO w/ drift demonstrates that incorporating policy drift effectively constrains the policy ratio within a reasonable range, thereby preventing training collapse.

On the other hand, while prior work suggests clipping may be unnecessary due to the low proportion of clipped ratios (Ahmadian et al., 2024; Chu et al., 2025), our findings suggest otherwise. Despite only ∼1% of ratios being clipped, training performance diverges significantly with and without clipping. Specifically, methods like PG and PGD—though stable without ratio terms—suffer from response length collapse, degenerating into trivial outputs (e.g., only emitting tokens like <think>) that exploit the format reward function without performing meaningful reasoning. This highlights the model's vulnerability to reward hacking, likely due to overly aggressive updates. These results reveal the necessity of the proximal policy updates.

## 4.3 IMPLEMENTATION

In this subsection, we design a practically implementable loss in per-token form based on the CPGD update formulation (Equation 2), aiming to strike a balance between theoretical rigor and empirical applicability. This practical loss is straightforward to be integrated into widely-used LLMs training frameworks like OpenRLHF (Hu et al., 2024) and veRL (Sheng et al., 2024):

$$\mathcal{J}_{\text{CPGD}}(\theta) = -\frac{1}{|\mathcal{D}|} \sum_{(\mathbf{x}, \{\mathbf{y}^{(k)}\}_{k=1}^K) \in \mathcal{D}} \frac{1}{\sum_{k=1}^K |\mathbf{y}^{(k)}|} \left[ \sum_{i=1}^{|\mathbf{y}^{(k)}|} \left( \Phi_\theta^i(\mathbf{x}, \mathbf{y}^{(k)}) - \alpha \cdot \mathcal{E}_{\theta_{old}, \theta}^i(\mathbf{x}, \mathbf{y}^{(k)}) \right) \right], \quad (3)$$

where

$$\Phi_\theta^i(\mathbf{x}, \mathbf{y}) := \min\left( \ln \frac{\pi_\theta(y_i|\mathbf{x}, \mathbf{y}_{<i})}{\pi_{\theta_{old}}(y_i|\mathbf{x}, \mathbf{y}_{<i})} \cdot A_\omega^{\text{CPGD}}(\mathbf{x}, \mathbf{y}), \text{clip}_{\ln(1-\epsilon_i)}^{\ln(1+\epsilon_i)}\left( \ln \frac{\pi_\theta(y_i|\mathbf{x}, \mathbf{y}_{<i})}{\pi_{\theta_{old}}(y_i|\mathbf{x}, \mathbf{y}_{<i})} \right) A_\omega^{\text{CPGD}}(\mathbf{x}, \mathbf{y}) \right),$$

$$A_\omega^{\text{CPGD}}(\mathbf{x}, \mathbf{y}^{(k)}) := \omega(\mathbf{x}) \cdot \left( \mathcal{R}_o(\mathbf{x}, \mathbf{y}^{(k)}) - \text{mean}\left( \{\mathcal{R}_o(\mathbf{x}, \mathbf{y}^{(k')})\}_{k'=1}^K \right) \right),$$

$$\mathcal{E}_{\theta_{old}, \theta}^i(\mathbf{x}, \mathbf{y}) := \min\left( \frac{\text{sg}(\pi_\theta(y_i|\mathbf{x}, \mathbf{y}_{<i}))}{\pi_{\theta_{old}}(y_i|\mathbf{x}, \mathbf{y}_{<i})} - 1, c \right) \cdot \ln \pi_\theta(y_i|\mathbf{x}, \mathbf{y}_{<i}).$$

Here, $\text{sg}(\cdot)$ denotes the operation that prevents gradient computation, $\omega(\mathbf{x})$ is a per-prompt weighting factor, and $c > 0$ is a constant. We provide the following clarifications regarding the differences between the theoretical update formulation (Equation 2) and the practical loss (Equation 3):

**(I) Policy optimization term:** In the theoretical update (Equation 2), the policy optimization term is written in the form of joint distribution. However, in the practical implementation (Equation 3), it is decomposed into token level using the decomposability of the logarithm function. Specifically, the clipping threshold $\epsilon_i$ can be set the same for all tokens, ensuring that each token shares the same clip range. Alternatively, a tight-to-loose schedule can be employed such as $\epsilon_i = \lambda \epsilon + (1-\lambda)\epsilon \cdot i/|\mathbf{y}^{(k)}|$, which assigns smaller thresholds to earlier tokens that usually have higher variance.

**(II) Policy drift:** Policy drift also leverages the decomposability of the logarithm function, but applies the following further transformations:

$$D_{\text{KL}}(\pi_{\theta_{old}}, \pi_\theta|\mathbf{x}) = \mathbb{E}_{\mathbf{y} \sim \pi_{\theta_{old}}(\cdot|\mathbf{x})}\left[ \ln \frac{\pi_{\theta_{old}}(\mathbf{y}|\mathbf{x})}{\pi_\theta(\mathbf{y}|\mathbf{x})} \right] = \mathbb{E}_{\mathbf{y} \sim \pi_{\theta_{old}}(\cdot|\mathbf{x})}\left[ \sum_{i=1}^{|\mathbf{y}|} \ln \frac{\pi_{\theta_{old}}(y_i|\mathbf{x}, \mathbf{y}_{<i})}{\pi_\theta(y_i|\mathbf{x}, \mathbf{y}_{<i})} \right] \quad (4)$$

$$= \mathbb{E}_{\mathbf{y} \sim \pi_{\theta_{old}}(\cdot|\mathbf{x})}\left[ \sum_{i=1}^{|\mathbf{y}|} \left( \frac{\pi_\theta(y_i|\mathbf{x}, \mathbf{y}_{<i})}{\pi_{\theta_{old}}(y_i|\mathbf{x}, \mathbf{y}_{<i})} - 1 - \ln \frac{\pi_\theta(y_i|\mathbf{x}, \mathbf{y}_{<i})}{\pi_{\theta_{old}}(y_i|\mathbf{x}, \mathbf{y}_{<i})} \right) \right]. \quad (5)$$

Equations 4 and 5 correspond to the $k_1$ and $k_3$ estimators of the KL divergence. However, both have drawbacks. The $k_1$ estimator yields a one-side gradient direction, regardless of how far the policy

has drifted, leading to wrong correction. The $k_3$ estimator provides a directionally adaptive gradient, but can become numerically unstable when the policy ratio is large:

$$\nabla_\theta \ln \frac{\pi_{\theta_{old}}(y_i|\mathbf{x}, \mathbf{y}_{<i})}{\pi_\theta(y_i|\mathbf{x}, \mathbf{y}_{<i})} = -\nabla_\theta \ln \pi_\theta(y_i|\mathbf{x}, \mathbf{y}_{<i}),$$

$$\nabla_\theta \left( \frac{\pi_\theta(y_i|\mathbf{x}, \mathbf{y}_{<i})}{\pi_{\theta_{old}}(y_i|\mathbf{x}, \mathbf{y}_{<i})} - 1 - \ln \frac{\pi_\theta(y_i|\mathbf{x}, \mathbf{y}_{<i})}{\pi_{\theta_{old}}(y_i|\mathbf{x}, \mathbf{y}_{<i})} \right) = \left( \frac{\pi_\theta(y_i|\mathbf{x}, \mathbf{y}_{<i})}{\pi_{\theta_{old}}(y_i|\mathbf{x}, \mathbf{y}_{<i})} - 1 \right) \nabla_\theta \ln \pi_\theta(y_i|\mathbf{x}, \mathbf{y}_{<i}).$$

To address this, we propose a clipped gradient variant of $k_3$ that retains its correctness of correction direction while improving stability. Specifically, our estimator $\mathcal{E}^i_{\theta_{old}, \theta}$ has the following gradient:

$$\nabla_\theta \mathcal{E}^i_{\theta_{old}, \theta}(\mathbf{x}, \mathbf{y}) = \min \left( \frac{\text{sg}(\pi_\theta(y_i|\mathbf{x}, \mathbf{y}_{<i}))}{\pi_{\theta_{old}}(y_i|\mathbf{x}, \mathbf{y}_{<i})} - 1, c \right) \cdot \nabla_\theta \ln \pi_\theta(y_i|\mathbf{x}, \mathbf{y}_{<i}).$$

This ensures that: (1) When the policy ratio is moderate, the behavior matches the $k_3$ estimator; (2) When the ratio exceeds the threshold $c + 1$, the gradient is capped but still points in the correct corrective direction. In summary, our estimator uniquely combines correct corrective direction and numerical stability, outperforming both $k_1$ and $k_3$ estimators in controlling policy drift effectively.

**(III) Weighted advantage:** In the view of the response level, each prompt can be viewed as a distinct task. Consequently, we can introduce a per-prompt weighting factor $\omega(\mathbf{x})$ to assign different levels of importance to different prompts. (1) *Equal weight*: when $\omega(\mathbf{x}) = 1$, $A^{\text{CPGD}}_\omega$ reduces to the original unweighted form. (2) *STD weight*: when $\omega(\mathbf{x}) = 1/\text{std}(\{\mathcal{R}(\mathbf{x}, \mathbf{y}^{(k)})\}_k)$, $A^{\text{CPGD}}_\omega$ is the same as $A^{\text{GRPO}}$. (3) *Clip-filter-like weight*: when $\omega(\mathbf{x}) = \min(c_\omega, \frac{\#\{\mathbf{x} \in \mathcal{D}\}}{\#\{\mathbf{x} \in \mathcal{D} | \text{std}(\{\mathcal{R}_o(\mathbf{x}, \mathbf{y}^{(k)})\}_k) \neq 0\}})$, $c_\omega > 0$, similar weighting strategies have also been explored in concurrent work (Chu et al., 2025), with an analogous effect to online filtering (Cui et al., 2025), amplifying the gradient contribution of samples with non-zero advantage.

## 5 EXPERIMENTS

### 5.1 EXPERIMENTS SETUP

**RL baselines, dataset, and implementation details.** We compare CPGD with several widely used RL algorithms, including GRPO (DeepSeek-AI et al., 2025), REINFORCE++ (Hu et al., 2025a) and RLOO (Ahmadian et al., 2024) on the MMK12 training dataset (Meng et al., 2025), which contains 15,616 multimodal math problems with verified answers. We use QwenVL2.5-7B as base models, and conduct experiments with **five** random seeds [1]. In Appendix C, we further provide supplementary results comparing CPGD and GRPO on **QwenVL2.5-32B**, **InternVL2.5-8B** and **Qwen3-8B** (**text-only**) to demonstrate the generality of our algorithms. Training is performed without reference constraints, and final performance is reported using the last checkpoint. Our rule-based reward consists of accuracy and format components: the former uses MathVerify to extract and compare answers, returning 1 or 0; the latter checks format compliance, returning 0.5 or 0. Details of hyperparameters and the system prompt are provided in Appendix C.

**Benchmarks, model baselines, and overall metric.** We evaluate all algorithms on six widely used benchmarks: MathVista (testmini) (Lu et al., 2024), MathVerse (testmini) (Zhang et al., 2024), MathVision (test) (Wang et al., 2024a), OlympiadBench (EN-OE split) (He et al., 2024), We-Math (Qiao et al., 2024) and MMK12 (Meng et al., 2025). See Appendix C for the details.

We also include several multimodal models as baselines. We evaluate open-source models of comparable model size, trained with various strategies, including QwenVL2.5 (Bai et al., 2023), InternVL2.5-MPO (Wang et al., 2024b), R1-OneVision (Yang et al., 2025), OpenVLThinker (Deng et al., 2025), and MM-Eureka (Meng et al., 2025), which collectively represent the average performance across the evaluated benchmarks. We further evaluate the leading closed-source models such as GPT-4o (Hurst et al., 2024) and OpenAI-o1 (OpenAI, 2024) to represent the most outstanding performance that the current state-of-the-art model can achieve on these benchmarks. Furthermore,

---

[1] Although in the field of LMs it is common to report results from a single random seed (due to high computational cost), we have run each set of experiments with five random seeds to ensure academic rigor and reproducibility.

to capture overall model performance across $N$ benchmarks, we define an *overall* metric by normalizing each score against a strong baseline, QwenVL2.5-7B: Overall $:= \frac{1}{N} \sum_{j=1}^{N} X_j / X_j^{\text{Qwen}}$, where $X_j$ and $X_j^{\text{Qwen}}$ are the model and baseline scores on benchmark $j$.

## 5.2 MAIN RESULTS

Table 1: Performance comparison of various 7B/8B models and leading closed-source models. Best mean in **bold**, second-best underlined (excl. OpenAI-o1/GPT-4o).

| Model | MathVista | MathVerse | MathVision | Olypamid | WeMath | MMK12 | Overall |
|---|---|---|---|---|---|---|---|
| **Leading models** | | | | | | | |
| GPT-4o | 63.8 | 50.2 | 30.4 | 35.0 | 68.8 | 49.9 | 1.16 |
| OpenAI-o1 | 73.9 | 57.0 | 60.3 | 68.0 | 98.7 | 73.9 | 1.83 |
| **Similar-size models** | | | | | | | |
| QwenVL2.5-7B | 68.2 | 47.9 | 25.4 | 20.2 | 62.1 | 53.6 | 1.00 |
| InternVL2.5-MPO-8B | 68.9 | 35.5 | 21.5 | 7.8 | 53.5 | 34.5 | 0.75 |
| R1-Onevision (7B) | 64.1 | 47.1 | 23.5 | 17.3 | 61.8 | 39.8 | 0.91 |
| OpenVLThinker (7B) | 70.2 | 47.9 | 25.3 | 20.1 | 64.3 | 60.6 | 1.03 |
| MM-Eureka (7B) | 73.0 | 50.3 | 26.9 | 20.1 | 66.1 | 64.5 | 1.07 |
| **Different RL algorithms on QwenVL2.5-7B** | | | | | | | |
| RLOO | 70.5±1.3 | 49.0±0.9 | 20.7±1.3 | 18.9±0.4 | 67.2±1.0 | 62.1±0.7 | 1.01±0.00 |
| REINFORCE++ | 63.8±0.9 | 46.1±0.7 | 18.9±0.4 | 18.7±0.6 | 66.6±0.6 | 64.7±0.3 | 0.98±0.01 |
| GRPO | 70.7±0.8 | 50.6±0.7 | 23.0±1.6 | 19.4±0.6 | 67.2±0.6 | 65.0±0.1 | 1.04±0.01 |
| *CPGD* (ours) | **73.8**±0.5 | **51.1**±0.7 | **27.0**±0.9 | **21.2**±0.4 | **68.0**±0.6 | **66.8**±0.8 | **1.10**±0.01 |

Table 1 presents a comprehensive comparison across multiple multimodal mathematical benchmarks. Closed-source models GPT-4o and OpenAI-o1 demonstrate strong performance across all tasks, with o1 achieving the highest scores overall, notably excelling on MathVision (60.3), Olypamid (68.0) and WeMath (98.7), establishing the current performance upper bound. Among similar-size open models, MM-Eureka shows competitive results. MM-Eureka achieves strong results on MathVista (73.0), MathVision (26.9) and a strong result on MMK12 (64.5). However, our proposed CPGD generally outperforms the similar-size baselines, achieving top or near-leading scores across all benchmarks, reflecting the effectiveness of our proposed RL algorithm.

We further analyze different RL algorithms under the same setting, including the base model, the training dataset, and the hyperparameters. Among the baseline methods, CPGD outperforms popular RL algorithms such as RLOO, REINFORCE++, and GRPO on benchmark tests, particularly on MathVista (73.8) and MathVision (27.0). Compared with the base model QwenVL2.5-7B, CPGD achieves an overall improvement of 10%. Notably, CPGD attains a 24.6% gain on the in-domain benchmark MMK12, and achieves 8.2% and 9.5% improvements on the out-of-distribution benchmarks MathVista and WeMath, respectively, further demonstrating its generalization capability.

In addition, we provide results in the Appendix C comparing CPGD with GRPO on **InternVL2.5-8B**, **QwenVL2.5-32B**, and **Qwen3-8B (text-only)**. These further experiments confirm the strong generalization ability of CPGD across different model backbones and task settings. Taken together, these results demonstrate that CPGD serves as a strong and robust alternative for RL in LM training.

## 5.3 ABLATION STUDY

**Component ablation.** We conduct ablation on key components of our method by comparing variants: PG (basic policy gradient), PGD (PG + policy drift), CPG (PG + clip mechanism), and CPGD. Results show that the clip mechanism plays the most critical role, as seen by the performance drop from CPG/CPGD to PG/PGD across nearly all benchmarks. This aligns with our observation in Section 4.2 that clipping mitigates the response length collapse issue, which otherwise can impair test-time computation and reasoning capabilities. In contrast, adding policy drift has a relatively smaller effect. This is because CPGD's objective lacks a potentially unstable importance-sampling ratio and already benefits from proximal updates via clipping, making policy drift mainly serve as a safeguard against excessive ratio deviation.

Table 2: Results of ablation studies. Best mean in **bold**, and * indicates no significant difference from best (bootstrap, 10,000 resamples, 5% level).

| Model | MathVista | MathVerse | MathVision | Olypamid | WeMath | MMK12 | Overall |
|---|---|---|---|---|---|---|---|
| *CPGD* (STD weight) | **73.8**±0.5 | 51.1±0.7 | **27.0**±0.9 | **21.2**±0.4 | 68.0±0.6* | 66.8±0.8* | **1.10**±0.01 |
| **Ablation study on the components (using STD weight)** | | | | | | | |
| PG | 67.4±0.7 | 41.3±0.9 | 21.4±0.8 | 9.1±0.8 | 57.7±0.7 | 63.8±1.8 | 0.88±0.01 |
| PGD | 65.8±1.5 | 41.7±0.6 | 20.9±0.7 | 8.8±1.3 | 57.5±0.6 | 66.4±0.7* | 0.87±0.01 |
| CPG | 71.6±1.8 | 52.4±2.0* | 24.3±2.6 | 20.8±0.8* | **69.4**±1.5 | 66.6±0.4* | 1.08±0.02* |
| **Ablation study on the weighting factor** | | | | | | | |
| unprocessed rewards | 68.9±0.5 | 41.0±0.7 | 21.2±0.6 | 3.5±0.5 | 59.1±0.5 | **66.9**±0.2 | 0.84±0.00 |
| equal weight | 72.2±0.8 | 50.8±0.4 | 23.5±2.7 | 20.1±0.4 | 67.1±0.5 | 66.1±0.5 | 1.06±0.02 |
| clip-filter-like weight | 73.1±0.6 | **52.6**±0.7 | 26.0±0.4 | 20.4±0.6 | 69.2±0.7* | 66.5±0.5* | 1.09±0.00* |
| **Ablation study on the reference constraint (using STD weight)** | | | | | | | |
| w/ reference constraint | 71.4±0.6 | 50.2±0.7 | 22.3±1.0 | 21.1±0.2* | 68.7±0.9* | 64.8±0.9 | 1.06±0.01 |

**Weighting factor ablation.** We further ablate different weighting strategies. We include a baseline using raw *unprocessed rewards* as advantages, which severely degrades performance. This confirms that subtracting the group mean is crucial for effective learning, which prevents over-penalization of all responses in the failure cases, which may otherwise trigger a *squeezing effect* (Ren & Sutherland, 2025), where the Softmax head shifts probability mass to unintended tokens. Both clip-filter-like and STD weighting outperform equal weighting by emphasizing samples with non-zero advantages. This targeted weighting encourages the model to focus more on informative training signals, thereby contributing to the improved performance.

**Reference constraint ablation.** Using a small weight of 0.001 still leads to a performance drop, while removing the reference constraint consistently improves results, indicating that such constraints may overly limit policy optimization (Yu et al., 2025; Liu et al., 2025; Hu et al., 2025b).

## 6  DISCUSSION ON IMPORTANCE SAMPLING

Importance sampling corrects distribution mismatch between the behavior and learned policies, improving sample efficiency. We omit the ratio to reduce variance, but do not recommend discarding it entirely. Our decision is based on two key observations: (1) the clipping fraction is only ~1% (Figure 1), and (2) we use a single PPO epoch. Thus, we argue that the importance-sampling ratio should be reintroduced when the clipping fraction is larger or multiple PPO epochs are applied:

$$A_\omega^{\text{CPGD}}(\mathbf{x}, \mathbf{y}) \leftarrow \mathcal{C}\left(\frac{\text{sg}(\pi_\theta(y_i|\mathbf{x}, \mathbf{y}_{<i}))}{\pi_{\theta_{old}}(y_i|\mathbf{x}, \mathbf{y}_{<i})}\right) A_\omega^{\text{CPGD}}(\mathbf{x}, \mathbf{y}).$$

Here, $\mathcal{C}(\cdot)$ denotes an arbitrary truncation function, such as $\text{clip}_{1-\epsilon}^{1+\epsilon}(\cdot)$. Compared to PPO, CPGD decouples the importance-sampling ratio from the gradient-carrying term, offering greater flexibility in designing and applying the ratio. See Appendix C for more discussion and related experiments.

In addition, we provide a discussion about the comparison between forward KL divergence and reverse KL divergence in Appendix D.

## 7  CONCLUSION

We identify a critical source of instability in existing RL methods for LMs: the use of asymmetric clipping on importance-sampling ratios, which can result in training collapse. To address this, we propose *CPGD*, a principled alternative that avoids direct dependence on policy ratios while enforcing proximal updates through the clip mechanism and policy drift. CPGD further incorporates a stable KL estimator and a weighted advantage strategy to improve learning robustness. Theoretically grounded and empirically validated, CPGD demonstrates superior stability and performance across multimodal math benchmarks, offering a strong and stable RL solution for training LMs.

## 8 ETHICS STATEMENT

This work adheres to the ICLR Code of Ethics. No human subjects or animal experimentation was involved in this study. The dataset used is open-source and complies with relevant usage guidelines. We avoided biases and discriminatory outcomes when utilizing the Qwen/QwenVL/InternVL series model. No personally identifiable information was used, and no experiments were conducted that could raise privacy or security concerns. We are committed to maintaining transparency and integrity throughout the research process.

## 9 REPRODUCIBILITY STATEMENT

We have made every effort to ensure the reproducibility of our results. Our code is available in the Supplementary Material to facilitate reproduction and verification. This paper provides detailed descriptions of experimental settings, including training procedures, model configurations, and hardware specifications. We also provide the scripts of experimental runs in the Supplementary Material to help others reproduce our experiments. Additionally, the training dataset and the QwenVL/Qwen/InternVL training model used are open-source, ensuring consistency and reproducibility of evaluation results. We believe these measures enable other researchers to reproduce our work and further advance the field.

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

# A THE USE OF LARGE LANGUAGE MODELS (LLMS)

LLMs were employed during the writing of this paper to polish the text and correct grammatical errors. The prompt used was: "Please detect and correct any grammatical errors in the following text, and polish it to enhance its academic expression. <text>"

# B PROOFS

## B.1 PROOF FOR PROPOSITION 1

**Proposition 2.** *Let $\theta_0$ be a parameter such that the importance-sampling ratio satisfies $\left|\frac{\pi_{\theta_0}(\mathbf{y}|\mathbf{x})}{\pi_{\theta_{old}}(\mathbf{y}|\mathbf{x})} - 1\right| = \epsilon$. Consider updating $\theta_0$ using either (i) the PPO-clip objective, resulting in parameter $\theta_1^{PPO}$, or (ii) the CPGD objective with $\alpha = 0$, yielding parameter $\theta_1^{CPG}$. Then, there exists a constant $\eta_{\max} > 0$ such that for any learning rate $\eta \in (0, \eta_{\max})$, the following inequality holds:*

$$\left|\frac{\pi_{\theta_1^{PPO}}(\mathbf{y}|\mathbf{x})}{\pi_{\theta_{old}}(\mathbf{y}|\mathbf{x})} - 1\right| > \left|\frac{\pi_{\theta_1^{CPG}}(\mathbf{y}|\mathbf{x})}{\pi_{\theta_{old}}(\mathbf{y}|\mathbf{x})} - 1\right| > \left|\frac{\pi_{\theta_0}(\mathbf{y}|\mathbf{x})}{\pi_{\theta_{old}}(\mathbf{y}|\mathbf{x})} - 1\right| = \epsilon.$$

*After one update step, both PPO and CPG increase the importance-sampling ratio deviation from the old policy, but PPO does so more aggressively than CPG.*

*Proof.* Consider $f(\eta) = \frac{\pi_{\theta_1^{\mathrm{CPG}}}(\mathbf{y}|\mathbf{x})}{\pi_{\theta_{old}}(\mathbf{y}|\mathbf{x})}$, where $\theta_1^{\mathrm{CPG}} = \theta_0 + \eta \nabla_\theta \hat{\mathcal{L}}_{\mathrm{CPG}}(\mathbf{x}, \mathbf{y}; \theta_0)$ is the single gradient ascent step on the empirical CPGD objective (Equation 2) without the policy drift term. The gradient of the objective takes the form:

$$\nabla_\theta \hat{\mathcal{L}}_{\mathrm{CPG}}(\mathbf{x}, \mathbf{y}; \theta) = A^{\mathrm{CPGD}}(\mathbf{x}, \mathbf{y}) \nabla_\theta \ln \pi_\theta(\mathbf{y}|\mathbf{x}).$$

Thus, for the case where $\frac{\pi_{\theta_0}(\mathbf{y}|\mathbf{x})}{\pi_{\theta_{old}}(\mathbf{y}|\mathbf{x})} = 1 + \epsilon$ and $A^{\mathrm{CPGD}}(\mathbf{x}, \mathbf{y}) > 0$, the directional derivative of $f$ at $\eta = 0$ satisfies:

$$f'(0) = \langle \frac{\nabla_\theta \pi_{\theta_0}(\mathbf{y}|\mathbf{x})}{\pi_{\theta_{old}}(\mathbf{y}|\mathbf{x})}, \nabla_\theta \hat{\mathcal{L}}_{\mathrm{CPG}}(\mathbf{x}; \theta_0) \rangle > 0.$$

Hence, there exists a constant $\eta_1 > 0$ such that for any $\eta \in (0, \eta_1)$, we have $f(\eta) > f(0)$. Similarly, when $\frac{\pi_{\theta_0}(\mathbf{y}|\mathbf{x})}{\pi_{\theta_{old}}(\mathbf{y}|\mathbf{x})} = 1 - \epsilon$ and $A^{\mathrm{CPGD}}(\mathbf{x}, \mathbf{y}) < 0$, there exists $\eta_2 > 0$ such that $f(\eta) < f(0)$ for any $\eta \in (0, \eta_2)$.

Therefore, for any $0 < \eta < \min(\eta_1, \eta_2)$, the following holds:

$$|\frac{\pi_{\theta_1^{\mathrm{CPG}}}(\mathbf{y}|\mathbf{x})}{\pi_{\theta_{old}}(\mathbf{y}|\mathbf{x})} - 1| > |\frac{\pi_{\theta_0}(\mathbf{y}|\mathbf{x})}{\pi_{\theta_{old}}(\mathbf{y}|\mathbf{x})} - 1| = \epsilon. \tag{6}$$

Next, define $g(\eta) = \frac{\pi_{\theta_1^{\mathrm{CPG}}}(\mathbf{y}|\mathbf{x})}{\pi_{\theta_{old}}(\mathbf{y}|\mathbf{x})} - \frac{\pi_{\theta_1^{\mathrm{PPO}}}(\mathbf{y}|\mathbf{x})}{\pi_{\theta_{old}}(\mathbf{y}|\mathbf{x})}$, where $\theta_1^{\mathrm{PPO}} = \theta_0 + \eta \nabla_\theta \hat{\mathcal{L}}_{\mathrm{PPO}}(\mathbf{x}, \mathbf{y}; \theta_0)$ and

$$\nabla_\theta \hat{\mathcal{L}}_{\mathrm{PPO}}(\mathbf{x}, \mathbf{y}; \theta) = A^{\mathrm{CPGD}}(\mathbf{x}, \mathbf{y}) \frac{\nabla_\theta \pi_\theta(\mathbf{y}|\mathbf{x})}{\pi_{\theta_{old}}(\mathbf{y}|\mathbf{x})}.$$

For the case where $\frac{\pi_{\theta_0}(\mathbf{y}|\mathbf{x})}{\pi_{\theta_{old}}(\mathbf{y}|\mathbf{x})} = 1 + \epsilon$ and $A^{\mathrm{CPGD}}(\mathbf{x}, \mathbf{y}) > 0$, we have:

$$g'(0) = \langle \frac{\nabla_\theta \pi_{\theta_0}(\mathbf{y}|\mathbf{x})}{\pi_{\theta_{old}}(\mathbf{y}|\mathbf{x})}, A^{\mathrm{CPGD}}(\mathbf{x}, \mathbf{y}) \cdot (1 - \frac{\pi_\theta(\mathbf{y}|\mathbf{x})}{\pi_{\theta_{old}}(\mathbf{y}|\mathbf{x})}) \cdot \nabla_\theta \ln \pi_\theta(\mathbf{y}|\mathbf{x}) \rangle < 0.$$

Hence, there exists a constant $\eta_3 > 0$ such that $g(\eta) < g(0)$ for any $\eta \in (0, \eta_3)$. Similarly, for the case where $\frac{\pi_{\theta_0}(\mathbf{y}|\mathbf{x})}{\pi_{\theta_{old}}(\mathbf{y}|\mathbf{x})} = 1 - \epsilon$ and $A^{\mathrm{CPGD}}(\mathbf{x}, \mathbf{y}) < 0$, there exists a constant $\eta_4 > 0$ such that $g(\eta) > g(0)$ for any $\eta \in (0, \eta_4)$.

Therefore, for any $0 < \eta < \min(\eta_3, \eta_4)$, we have

$$|\frac{\pi_{\theta_1^{\mathrm{PPO}}}(\mathbf{y}|\mathbf{x})}{\pi_{\theta_{old}}(\mathbf{y}|\mathbf{x})} - 1| > |\frac{\pi_{\theta_1^{\mathrm{CPG}}}(\mathbf{y}|\mathbf{x})}{\pi_{\theta_{old}}(\mathbf{y}|\mathbf{x})} - 1|. \tag{7}$$

Therefore, by letting $\eta_{\max} = \min(\eta_1, \eta_2, \eta_3, \eta_4)$, the proof is complete. $\square$

## B.2 PROOF FOR THEOREM 1

**Theorem 2.** *Let $\{\pi_{\theta_k}\}_{k=0}^\infty$ denote the sequence of policies generated by the CPGD update rule: $\theta_{k+1} = \arg\max_\theta \mathcal{L}_{CPGD}(\theta; \theta_{old})$ where the advantage function is competed as $A^{CPGD}(\mathbf{x}, \mathbf{y}) = \mathcal{R}_o(\mathbf{x}, \mathbf{y})$. Then, $\pi_{\theta_{k+1}}$ is better than $\pi_{\theta_k}$, i.e., $\eta(\theta_{k+1}) \geq \eta(\theta_k)$, where $\eta(\theta) := \mathbb{E}_{\mathbf{x} \sim \mathcal{D}, \mathbf{y} \sim \pi_{\theta_{old}}(\cdot|\mathbf{x})}[\mathcal{R}_o(\mathbf{x}, \mathbf{y})]$.*

*Proof.* First, denote $\mathcal{L}_{\mathrm{CPGD}}(\theta; \theta_k) = \mathbb{E}_{\mathbf{x} \sim \mathcal{D}}[g(\theta; \theta_k, \mathbf{x})]$, and rewrite $g$ as

$$g(\theta; \theta_k, \mathbf{x}) = \mathbb{E}_{\mathbf{y} \sim \pi_{\theta_k}(\cdot|\mathbf{x})}\left[ \mathcal{R}_o(\mathbf{x}, \mathbf{y}) \ln \frac{\pi_\theta(\mathbf{y}|\mathbf{x})}{\pi_{\theta_k}(\mathbf{y}|\mathbf{x})} \right] - \alpha D_{\mathrm{KL}}(\pi_{\theta_k}, \pi_\theta|\mathbf{x})$$

$$- \mathbb{E}_{\mathbf{y} \sim \pi_{\theta_k}(\cdot|\mathbf{x})}\left[ \mathrm{ReLU}\left( \left[ \ln \frac{\pi_\theta(\mathbf{y}|\mathbf{x})}{\pi_{\theta_k}(\mathbf{y}|\mathbf{x})} - \mathrm{clip}\left( \ln \frac{\pi_\theta(\mathbf{y}|\mathbf{x})}{\pi_{\theta_k}(\mathbf{y}|\mathbf{x})}, \ln(1 - \epsilon), \ln(1 + \epsilon) \right) \right] \mathcal{R}_o(\mathbf{x}, \mathbf{y}) \right) \right],$$

which is obtained by the following observation:

$$\mathbb{E}_{\mathbf{y}\sim\pi_{\theta_k}(\cdot|\mathbf{x})}\Big[\min\Big\{\mathcal{R}_o(\mathbf{x},\mathbf{y})\ln\frac{\pi_\theta(\mathbf{y}|\mathbf{x})}{\pi_{\theta_k}(\mathbf{y}|\mathbf{x})},\mathcal{R}_o(\mathbf{x},\mathbf{y})\mathrm{clip}_{\ln(1-\epsilon)}^{\ln(1+\epsilon)}\Big(\ln\frac{\pi_\theta(\mathbf{y}|\mathbf{x})}{\pi_{\theta_k}(\mathbf{y}|\mathbf{x})}\Big)\Big\}\Big]$$

$$=\mathbb{E}_{\mathbf{y}\sim\pi_{\theta_k}(\cdot|\mathbf{x})}\Big[\mathcal{R}_o(\mathbf{x},\mathbf{y})\ln\frac{\pi_\theta(\mathbf{y}|\mathbf{x})}{\pi_{\theta_k}(\mathbf{y}|\mathbf{x})}\Big]-\mathbb{E}_{\mathbf{y}\sim\pi_{\theta_k}(\cdot|\mathbf{x})}\Big[\mathcal{R}_o(\mathbf{x},\mathbf{y})\ln\frac{\pi_\theta(\mathbf{y}|\mathbf{x})}{\pi_{\theta_k}(\mathbf{y}|\mathbf{x})}$$

$$-\min\Big\{\mathcal{R}_o(\mathbf{x},\mathbf{y})\ln\frac{\pi_\theta(\mathbf{y}|\mathbf{x})}{\pi_{\theta_k}(\mathbf{y}|\mathbf{x})},\mathcal{R}_o(\mathbf{x},\mathbf{y})\mathrm{clip}_{\ln(1-\epsilon)}^{\ln(1+\epsilon)}\Big(\ln\frac{\pi_\theta(\mathbf{y}|\mathbf{x})}{\pi_{\theta_k}(\mathbf{y}|\mathbf{x})}\Big)\Big\}\Big]$$

$$=\mathbb{E}_{\mathbf{y}\sim\pi_{\theta_k}(\cdot|\mathbf{x})}\Big[\mathcal{R}_o(\mathbf{x},\mathbf{y})\ln\frac{\pi_\theta(\mathbf{y}|\mathbf{x})}{\pi_{\theta_k}(\mathbf{y}|\mathbf{x})}\Big]-\mathbb{E}_{\mathbf{y}\sim\pi_{\theta_k}(\cdot|\mathbf{x})}\Big[\mathcal{R}_o(\mathbf{x},\mathbf{y})\ln\frac{\pi_\theta(\mathbf{y}|\mathbf{x})}{\pi_{\theta_k}(\mathbf{y}|\mathbf{x})}$$

$$-\max\Big\{-\mathcal{R}_o(\mathbf{x},\mathbf{y})\ln\frac{\pi_\theta(\mathbf{y}|\mathbf{x})}{\pi_{\theta_k}(\mathbf{y}|\mathbf{x})},-\mathcal{R}_o(\mathbf{x},\mathbf{y})\mathrm{clip}_{\ln(1-\epsilon)}^{\ln(1+\epsilon)}\Big(\ln\frac{\pi_\theta(\mathbf{y}|\mathbf{x})}{\pi_{\theta_k}(\mathbf{y}|\mathbf{x})}\Big)\Big\}\Big]$$

$$=\mathbb{E}_{\mathbf{y}\sim\pi_{\theta_k}(\cdot|\mathbf{x})}\Big[\mathcal{R}_o(\mathbf{x},\mathbf{y})\ln\frac{\pi_\theta(\mathbf{y}|\mathbf{x})}{\pi_{\theta_k}(\mathbf{y}|\mathbf{x})}\Big]$$

$$-\mathbb{E}_{\mathbf{y}\sim\pi_{\theta_k}(\cdot|\mathbf{x})}\Big[\max\Big\{0,\mathcal{R}_o(\mathbf{x},\mathbf{y})\Big(\ln\frac{\pi_\theta(\mathbf{y}|\mathbf{x})}{\pi_{\theta_k}(\mathbf{y}|\mathbf{x})}-\mathrm{clip}_{\ln(1-\epsilon)}^{\ln(1+\epsilon)}\Big(\ln\frac{\pi_\theta(\mathbf{y}|\mathbf{x})}{\pi_{\theta_k}(\mathbf{y}|\mathbf{x})}\Big)\Big)\Big\}\Big].$$

Here, we omit the baseline $\mathbb{E}_{\mathbf{y}\sim\pi_{\theta_k}(\cdot|\mathbf{x})}[\mathcal{R}_o(\mathbf{x},\mathbf{y})]$. Then, denoting $\theta_{k+1}$ the point such that $\mathcal{L}_{\mathrm{CPGD}}(\theta_{k+1};\theta_k)\geq\mathcal{L}_{\mathrm{CPGD}}(\theta_k;\theta_k)$, we obtain

$$\mathbb{E}_{\mathbf{y}\sim\pi_{\theta_{k+1}}(\cdot|\mathbf{x})}\Big[\mathcal{R}_o(\mathbf{x},\mathbf{y})\Big]-\mathbb{E}_{\mathbf{y}\sim\pi_{\theta_k}(\cdot|\mathbf{x})}\Big[\mathcal{R}_o(\mathbf{x},\mathbf{y})\Big]$$

$$=\mathbb{E}_{\mathbf{y}\sim\pi_{\theta_k}(\cdot|\mathbf{x})}\Big[\Big(\frac{\pi_{\theta_{k+1}}(\mathbf{y}|\mathbf{x})}{\pi_{\theta_k}(\mathbf{y}|\mathbf{x})}-1\Big)\mathcal{R}_o(\mathbf{x},\mathbf{y})\Big]$$

$$\geq\mathbb{E}_{\mathbf{y}\sim\pi_{\theta_k}(\cdot|\mathbf{x})}\Big[\ln\frac{\pi_{\theta_{k+1}}(\mathbf{y}|\mathbf{x})}{\pi_{\theta_k}(\mathbf{y}|\mathbf{x})}\cdot\mathcal{R}_o(\mathbf{x},\mathbf{y})\Big]$$

$$=g(\theta_{k+1};\theta_k,\mathbf{x})-g(\theta_k;\theta_k,\mathbf{x})+\alpha D_{\mathrm{KL}}(\pi_{\theta_k},\pi_{\theta_{k+1}}|\mathbf{x})$$

$$+\mathbb{E}_{\mathbf{y}\sim\pi_{\theta_k}(\cdot|\mathbf{x})}\Big[\mathrm{ReLU}\Big(\big[\ln\frac{\pi_{\theta_{k+1}}(\mathbf{y}|\mathbf{x})}{\pi_{\theta_k}(\mathbf{y}|\mathbf{x})}-\mathrm{clip}\big(\ln\frac{\pi_{\theta_{k+1}}(\mathbf{y}|\mathbf{x})}{\pi_{\theta_k}(\mathbf{y}|\mathbf{x})},\ln(1-\epsilon),\ln(1+\epsilon)\big)\big]\mathcal{R}_o(\mathbf{x},\mathbf{y})\Big)\Big].$$

Denoting the overall expected return by $\eta(\pi_\theta)=\mathbb{E}_{\mathbf{x}\sim\mathcal{D},\mathbf{y}\sim\pi_\theta(\cdot|\mathbf{x})}\Big[\mathcal{R}_o(\mathbf{x},\mathbf{y})\Big]$, we integrate over $\mathbf{x}$ to conclude

$$\eta(\pi_{\theta_{k+1}})-\eta(\pi_{\theta_k})\geq\alpha\mathbb{E}_{\mathbf{x}\sim\mathcal{D}}\Big[D_{\mathrm{KL}}(\pi_{\theta_k},\pi_{\theta_{k+1}}|\mathbf{x})\Big]\overset{\text{Pinsker inequality}}{\geq}\frac{\alpha}{2}\|\pi_{\theta_{k+1}}-\pi_{\theta_k}\|_1^2.$$

Therefore, we have $\eta(\theta_{k+1})\geq\eta(\theta_k)$.

$\square$

## C  DETAILS OF EXPERIMENTS

### C.1  PROMPT SETTING

We follow the prompt format from DeepSeek-R1, where reasoning steps and final answers are explicitly marked using `<think>` and `<answer>` tags, respectively. The full prompt template is provided in Table 3.

### C.2  HYPERPARAMETERS

For all experiments, we use the same hyperparameters: rollout and training batch sizes of 128, 8 sampled responses per prompt (temperature 1.0), a learning rate of $1e-6$, one PPO epoch, and five training episodes. No reference policy constraint is applied during training, final performance is reported using the last checkpoint, and each run requires approximately 60 hours of computation on 8 H100 GPUs.

Table 3: Prompt setting.

**SYSTEM:** Solve the question. The user asks a question, and you solves it. You first thinks about the reasoning process in the mind and then provides the user with the answer. The answer is in latex format and wrapped in $...$. The final answer must be wrapped using the \boxed{} command. The reasoning process and answer are enclosed within <think></think> and <answer></answer> tags, respectively, i.e., <think>Since $1 + 1 = 2$, so the answer is 2. </think><answer>The answer is $\boxed{2}$ </answer>, which means the final answer assistant's output should start with <answer> and end with </answer>.
**USER:** <image>{{question}}

## C.3 DETAILS OF BENCHMARKS

We evaluate all algorithms on six widely used benchmarks: MathVista (testmini) (Lu et al., 2024), MathVerse (testmini) (Zhang et al., 2024), MathVision (test) (Wang et al., 2024a), OlympiadBench (EN-OE split) (He et al., 2024), WeMath (Qiao et al., 2024) and MMK12 (Meng et al., 2025). See Appendix C for the details of benchmarks. MathVista covers visual QA, logic, algebra, and geometry; MathVerse focuses on mathematically grounded visual understanding; and MathVision extends to abstract visual reasoning. OlympiadBench targets graduate-level competition problems, while WeMath enables fine-grained diagnostic analysis via hierarchically annotated tasks. MMK12 provides 500 multiple-choice questions per subject across math, physics, chemistry, and biology for cross-domain performance evaluation.

## C.4 ADDITION EXPERIMENT ON OTHER MODEL BACKBONES

Table 4: Comparisons of CPGD and GRPO on Internvl2.5 and QwenVL2.5-32B across all benchmarks.

| Model | MathVista | MathVerse | MathVision | Olypamid | WeMath | MMK12 | Overall |
|---|---|---|---|---|---|---|---|
| InternVL2.5 | 64.4 | 39.5 | 15.8 | 12.3 | 49.4 | 46.5 | 1.00 |
| InternVL2.5-GRPO | 66.8±0.6 | **41.1**±0.7 | 20.1±0.5 | 9.9±0.5 | 53.8±0.5* | 48.2±0.4 | 1.05±0.01 |
| InternVL2.5-CPGD | **68.8**±0.6 | 41.0±0.5* | **22.2**±0.7 | **13.3**±0.2 | **54.0**±0.3 | **49.2**±0.3 | **1.12**±0.01 |
| QwenVL2.5-32B | 71.7 | 49.9 | 40.1 | 30.0 | 69.1 | 66.8 | 1.00 |
| QwenVL2.5-32B-GRPO | 74.0±0.3 | 55.9±0.6 | 30.6±1.1* | 35.7±0.6 | 71.4±1.2 | 73.1±1.1 | 1.04±0.00 |
| QwenVL2.5-32B-CPGD | **75.5**±0.3 | **58.0**±0.5 | **31.8**±0.4 | **40.9**±0.3 | **74.2**±0.5 | **76.1**±0.3 | **1.10**±0.00 |

Table 5: Comparisons of CPGD and GRPO on QwenVL3-8B across all benchmarks (avg@8).

| Model | AIME2024 | AIME2025 | MATH-500 | Overall |
|---|---|---|---|---|
| Qwen3-8B | 10.5 | 10.4 | 60.1 | 1.00 |
| Qwen3-8B-GRPO | 23.6±0.6 | 22.5±1.2 | 72.4±0.9 | 1.87±0.05 |
| Qwen3-8B-CPGD | **28.4**±0.4 | **26.2**±0.9 | **75.6**±0.2 | **2.16**±0.02 |

Tables 4 and 5 present detailed comparisons for GRPO and CPGD on on InternVL2.5-8B, QwenVL2.5-32B, and Qwen3-8B (text-only). For the multimodal experiments, the training pipeline and hyperparameters are kept exactly the same as those used on QwenVL2.5-8B. For Qwen3-8B, we instead adopt the following hyperparameters: train batch size of 2048, rollout batch size of 512, and 16 responses per prompt (temperature 1.0), a learning rate of $1e-6$, one PPO epoch, and five training episodes. The train dataset we use is DAPO-17k-math (Yu et al., 2025). Furthermore, since Qwen3-8B demonstrates strong instruction-following ability, we only apply the MathVerify-based accuracy reward without using the format reward.

Table 6: Results of ablation studies. Top performer is in **bold** and second-best is underlined.

| Model | MathVista | MathVerse | MathVision | Olypamid | WeMath | MMK12 |
|---|---|---|---|---|---|---|
| CPGD | **73.8**±0.5 | 51.1±0.7* | **27.0**±0.9 | **21.2**±0.4 | **68.0**±0.6 | **66.8**±0.8 |
| **Ablation study on the importance-sampling (IS) ratio (using STD weight)** | | | | | | |
| CPGD w/ global clip IS | 73.2±0.3 | **51.5**±0.6 | 26.1±0.6 | 21.0±0.4* | 67.5±0.2 | 65.5±0.4 |
| CPGD w/ dual clip IS | 71.4±0.1 | 49.6±0.1 | 25.9±0.7 | 20.4±0.2 | 65.7±0.4 | 64.5±0.4 |

## C.5 ADDITIONAL EXPERIMENT ABOUT IMPORTANCE-SAMPLING RATIO

In Section 6, we reintroduce the importance sampling ratio into CPGD, formulated as:

$$A_\omega^{\text{CPGD}}(\mathbf{x}, \mathbf{y}) \leftarrow \mathcal{C}\Big(\frac{\text{sg}(\pi_\theta(y_i|\mathbf{x}, \mathbf{y}_{<i}))}{\pi_{\theta_{old}}(y_i|\mathbf{x}, \mathbf{y}_{<i})}\Big) A_\omega^{\text{CPGD}}(\mathbf{x}, \mathbf{y}),$$

where $\mathcal{C}(\cdot)$ denotes an arbitrary truncation function, used to control variance by bounding the importance weights. We evaluate two specific forms of $\mathcal{C}(\cdot)$:

$$\text{dual clip:} \quad \mathcal{C}\Big(\frac{\pi_\theta(y_i|\mathbf{x}, \mathbf{y}_{<i})}{\pi_{\theta_{old}}(y_i|\mathbf{x}, \mathbf{y}_{<i})}\Big) = \text{clip}_0^{1+\epsilon}\Big(\frac{\pi_\theta(y_i|\mathbf{x}, \mathbf{y}_{<i})}{\pi_{\theta_{old}}(y_i|\mathbf{x}, \mathbf{y}_{<i})}\Big) \cdot \mathbf{1}_{A_\omega^{\text{CPGD}}(\mathbf{x}, \mathbf{y}) \geq 0}$$

$$+ \text{clip}_{1-\epsilon}^c\Big(\frac{\pi_\theta(y_i|\mathbf{x}, \mathbf{y}_{<i})}{\pi_{\theta_{old}}(y_i|\mathbf{x}, \mathbf{y}_{<i})}\Big) \cdot \mathbf{1}_{A_\omega^{\text{CPGD}}(\mathbf{x}, \mathbf{y}) < 0},$$

$$\text{global clip:} \quad \mathcal{C}\Big(\frac{\pi_\theta(y_i|\mathbf{x}, \mathbf{y}_{<i})}{\pi_{\theta_{old}}(y_i|\mathbf{x}, \mathbf{y}_{<i})}\Big) = \text{clip}_{1-\epsilon}^{1+\epsilon}\Big(\frac{\pi_\theta(y_i|\mathbf{x}, \mathbf{y}_{<i})}{\pi_{\theta_{old}}(y_i|\mathbf{x}, \mathbf{y}_{<i})}\Big).$$

The introduction of the dual clip function enables CPGD to share nearly identical gradients with PPO with dual clip mechanism—except in cases where the advantage is negative and the importance-sampling ratio exceeds $c$. In contrast, the global clip function constrains all policy ratios strictly within the range $[1 - \epsilon, 1 + \epsilon]$. We empirically compare these variants and report their performance in Table C.5. Methods employing a global clip function achieve performance comparable to those omitting the importance-sampling ratio, likely due to the stricter truncation applied. In contrast, approaches using a dual clip function exhibit notable performance degradation. In particular, CPG without policy drift suffers from training collapse, consistent with our findings in Section 4.2. These results indicate that more stable integration of the importance-sampling ratio remains an open research problem.

# D DISCUSSION

## D.1 FORWARD KL DIVERGENCE VS. REVERSE KL DIVERGENCE

Our policy drift is based on the *forward KL divergence* $D_{\text{KL}}(\pi_{old}, \pi)$, which is also used in PPO-KL (Schulman et al., 2017). However, our approach differs fundamentally in how this KL is estimated and applied. PPO-KL typically uses the $k_1$ estimator or a better $k_3$ estimator, while we introduce a novel gradient-based estimator (Section 4.3) that offers both correct corrective gradients and numerical stability, overcoming the limitations of existing estimators like $k_1$ (incorrect gradient direction) and $k_3$ (instability).

*Reverse KL divergence* $D_{\text{KL}}(\pi, \pi_{old})$ is more commonly used in related work due to its connection to mirror descent and stronger convergence guarantees (Geist et al., 2019; Shani et al., 2020). Although these two KL forms are different in how they are calculated, they often lead to similar results in practice (Hsu et al., 2020). Their gradient difference is typically small during training, especially when the policy ratio is close to 1, which is common in stable learning regimes:

$$\nabla_\theta D_{\text{KL}}(\pi_\theta, \pi_{\theta_{old}}|\mathbf{x}) - \nabla_\theta D_{\text{KL}}(\pi_{\theta_{old}}, \pi_\theta|\mathbf{x}) \approx \mathbb{E}_{\mathbf{y} \sim \pi_{\theta_{old}}(\cdot|\mathbf{x})}\Big[\frac{1}{2}\Big(\frac{\pi_\theta(\mathbf{y}|\mathbf{x})}{\pi_{\theta_{old}}(\mathbf{y}|\mathbf{x})} - 1\Big)^2 \nabla_\theta \ln \pi_\theta(\mathbf{y}|\mathbf{x})\Big].$$

This approximation holds because $x \ln x \approx x - 1 + \frac{1}{2}(x-1)^2$ when $x$ is close to 1. Despite their similarity, we prefer forward KL for two main reasons: (1) It avoids importance sampling, which reverse KL requires; and (2) It can be cleanly split into per-token terms (see Equation 5), which is not possible with reverse KL due to the importance weights.

# E    LIMITATIONS

While this work introduces a stable and effective RL method for LMs training, it has several limitations: (1) For the weighted advantage component, we conducted only preliminary experiments and did not thoroughly explore the impact of different weighting factors. Our results suggest that using non-uniform weights yields better performance than trivial equal weighting, but further investigation is needed. (2) Our study focuses on on-policy training; we leave off-policy settings—where importance sampling is typically required—for future work. Ensuring training stability in the presence of importance sampling remains an open question.

