# OpenReview forum: "CPGD: Toward Stable Reinforcement Learning for Language Models"
_ICLR.cc/2026/Conference — ICLR 2026 Conference Withdrawn Submission_

### Official Review · Reviewer_twDn · 2025-10-15

**Soundness:** 2
**Presentation:** 3
**Contribution:** 1
**Rating:** 2
**Confidence:** 4

**Summary:**

This paper proposes CPGD (Clipped Policy Gradient Optimization with Policy Drift), a reinforcement learning algorithm designed to address the pervasive training collapse observed in current rule-based reward training of language models (LMs). The authors argue that existing methods—such as GRPO, REINFORCE++, and RLOO—suffer from excessive policy updates due to the direct use of the importance-sampling ratio in their loss functions. In particular, under negative advantages, the standard one-sided clipping mechanism (e.g., PPO-clip) fails to effectively constrain policy drift, leading to training instability or even catastrophic collapse.

To mitigate this issue, CPGD introduces the following key design choices:

- REINFORCE-style loss: Instead of using the raw importance-sampling ratio, CPGD employs the log-ratio form (i.e., $\ln \frac{\pi_\theta}{\pi_{\text{old}}}$), which avoids high-variance gradient estimates.
- Clipping on log-ratio: A clipping mechanism is applied directly to the log-ratio—i.e., $\text{clip}\big(\ln \frac{\pi_\theta}{\pi_{\text{old}}}\big)$—to bound policy updates more symmetrically and robustly.
- Forward KL regularization: A policy drift penalty based on forward KL divergence ($D_{\text{KL}}(\pi_{\text{old}} \| \pi_\theta)$) is incorporated, along with a novel clipped-gradient KL estimator that preserves the correct gradient direction (as in the k3 estimator) while ensuring numerical stability.
- Weighted advantages: Sample efficiency is enhanced by dynamically reweighting advantages per prompt, emphasizing informative training signals.

The authors validate CPGD on multiple multimodal mathematical reasoning benchmarks—including MMK12 and MathVista—and report substantial performance gains over strong baselines such as GRPO. The paper further supports its design with theoretical analysis (monotonic improvement guarantee) and comprehensive ablation studies.

**Strengths:**

- Accurate problem identification: The paper clearly pinpoints a critical yet previously underappreciated instability issue in current rule-based reinforcement learning methods: the lack of effective constraints on the importance-sampling ratio under negative advantages, which can lead to training collapse. This observation is practically valuable and aligns with empirical challenges faced in real-world LM fine-tuning.

- Comprehensive engineering implementation: The authors provide thorough implementation details, including hyperparameter configurations, ablation studies, and reproducibility instructions. The experimental design is rigorous—using five random seeds—and greatly enhances the credibility and reproducibility of the reported results.

**Weaknesses:**

- Severely limited algorithmic novelty:
  - The clipping mechanism is not novel: Applying clipping to the log-ratio (i.e., $\text{clip}(\ln \frac{\pi_\theta}{\pi_{\text{old}}})$) rather than the importance-sampling ratio itself is still a variant of the standard PPO-clip. More critically, the so-called “dual clip”—which imposes an upper bound on the ratio when the advantage is negative—was already proposed in PPO[1] and is implemented as a standard feature in widely used libraries such as Hugging Face’s TRL[2]. The paper fails to adequately cite or discuss these existing engineering practices and instead presents them as a new discovery. **So is the only novelty just adding a log to the ratio and then truncating it?**
  - The KL estimator lacks theoretical innovation: The proposed “clipped k3” estimator is essentially gradient clipping applied to importance weights—a well-established technique in off-policy RL and importance-sampling variance control. The paper provides no theoretical analysis showing improvements in bias, variance, or convergence guarantees over existing estimators. Merely combining “correct gradient direction” with “numerical stability” does not constitute a novel algorithmic contribution.
  - The overall framework follows a classical paradigm: CPGD can be succinctly described as REINFORCE loss + clipping + KL regularization, effectively a hybrid of REINFORCE and PPO. It introduces no new optimization principle, objective function, or theoretical foundation beyond existing proximal policy optimization methods.

- Inadequate coverage of related work:
  - The claim that “none of these approaches focus on the training instability issue” is inaccurate. Open-source RL frameworks such as TRL, veRL, and OpenRLHF already incorporate numerous stability-enhancing techniques—including dual clipping, ratio clipping, and adaptive KL penalties—that directly address the very instability the paper highlights. These practical advances are overlooked.
  - The paper does not include comparisons against classical KL-regularized baselines such as PPO-KL, adaptive KL penalty, or mirror descent with reverse KL, which weakens the empirical evaluation and overstates CPGD’s novelty.

- Theoretical analysis is underdeveloped:
  - Theorem 1 relies on strong simplifying assumptions—most notably setting the advantage function to the raw reward ($A^{\text{CPGD}}(x, y) = R_o(x, y)$), thereby ignoring baselines.

  - Proposition 1 only analyzes the deviation of the importance-sampling ratio after a single update step. While it shows that CPGD induces smaller policy shifts than PPO in one step, it does not explain or guarantee long-term training stability, which is the central claim of the paper.

References:

[1] Proximal Policy Optimization Algorithms, 2017.

[2] https://github.com/huggingface/trl

**Questions:**

1. On novelty: The paper emphasizes that  “dual clipping” is a novel contribution. However, the PPOTrainer in the widely used TRL library already supports similar behavior through configurable cliprange and custom advantage processing. Could the authors clearly articulate the essential algorithmic distinction between CPGD and these existing engineering implementations, and justify why this constitutes a genuine algorithmic innovation rather than a reparameterization or repackaging of known practices?

2. On experimental setup: All experiments in the paper are conducted without a reference policy constraint (i.e., no π
ref KL penalty), whereas methods like GRPO typically include such a constraint to prevent excessive policy drift from the initial model. Does this imply that part of CPGD’s performance gain stems not from its core design, but from the removal of an overly restrictive reference KL term? To ensure a fair comparison, could the authors provide results where both CPGD and GRPO are evaluated under identical conditions—either both with or both without the reference constraint?

---

### Official Review · Reviewer_mLot · 2025-10-20

**Soundness:** 3
**Presentation:** 3
**Contribution:** 3
**Rating:** 6
**Confidence:** 4

**Summary:**

This paper addresses the significant training instability and collapse observed in existing rule-based reinforcement learning (RL) methods like GRPO and REINFORCE++ when applied to language models, an issue the authors attribute to the use of importance-sampling ratios and improper clipping in the loss function. To solve this, the authors propose CPGD (Clipped Policy Gradient Optimization with Policy Drift), a novel algorithm designed for stable policy learning. CPGD replaces the standard PPO-clip loss with the REINFORCE loss to avoid direct reliance on unstable policy ratios , while simultaneously enforcing proximal updates through two key mechanisms: a policy drift constraint based on KL divergence to regularize updates, and a clip mechanism applied to the logarithm of the importance-sampling ratio. The authors provide theoretical justification for CPGD and demonstrate empirically that it mitigates the instability of prior methods, leading to significantly improved performance and training stability on reasoning benchmarks.

**Strengths:**

1. This paper tackles the critical and practical problem of training instability, which frequently leads to training collapse in existing rule-based reinforcement learning methods used for post-training language models.
2. The proposed CPGD algorithm is well-supported by theoretical analysis, including a proof of its monotonic improvement property and a justification for how it mitigates the policy drift amplified by standard PPO-clip objectives.
3. The method's effectiveness and superior generalization are rigorously demonstrated through empirical comparisons on multiple model backbones (including text-only and multimodal) and across several standard reasoning benchmarks, showing consistent performance gains over existing RL baselines.

**Weaknesses:**

1. The experimental comparison is limited to RLOO, REINFORCE++, and GRPO, failing to benchmark CPGD against the more recent and concurrent GRPO variants like DAPO.
2. The "Leading models" used for upper-bound comparison, GPT-4o and OpenAI-01, are from 2024, and may be considered dated by the ICLR 2026 review period, potentially misrepresenting the current state-of-the-art.
3. Some designs in DAPO seems not conflict with the designs in CPGD. Is it possible to integrate the designs of CPGD with existing designs in other algorithms?

**Questions:**

Please see paper weaknesses.

---

### Official Review · Reviewer_angP · 2025-11-01

**Soundness:** 2
**Presentation:** 3
**Contribution:** 2
**Rating:** 6
**Confidence:** 4

**Summary:**

The paper focuses on the problem of training instability for modern LLM RL algorithms such as PPO, GRPO, REINFORCE++ etc mainly due to improper policy clipping factor which leads to training collapse. To remidiate these instabilities, authors propose Clipped Policy Gradient Optimization with Policy Drift (CPGD). At a high level, their method makes peculiar alterations to GRPO Clipped objective and KL penalty.
1. they replace the ratios and clipped ratios within GRPO objective with log ratios and clipped log ratios
2. they experiment with different normalization denominators for the grouped advantage calculation - equal weight, std weight, clipped-filter like weight. They find std weigh, which is standard in GRPO works best.
3. they add a clipped version of kl3 (forward kl version instead of reverse kl) which they call as policy drift

In experiments over multiple vision and text dataset with 2 vision and 1 text model, their method shows some improvement over the GRPO baseline.

**Strengths:**

- new interesting clipping modifications to the classic GRPO objective motivated by the need to find more stable learning
- some theoretical analysis to prove their objective's increase in importance sampling ratio is necessarily lower than PPO/GRPO
- experiments on both text and vision domain.

**Weaknesses:**

- Moving away from importance sampling ratios to log importance sampling ratio seems like an arbitrary choice which doesn't necessarily derivable from the deriviation of off-policy policy gradient objective. Overall, their objective is a special weighted verison (w/ clipping) of REINFORCE with grouped advantage from GRPO.
- From the analysis in Figure 1, the conclusion of instability of other algorithms over CPGD seems more of an artifact of hyperparameter choice of the experiment (fixed lr 1e-6 for all methods). Experimented methods like RLOO and GRPO show clipped fraction blowup in this setting but other papers (Ahmadian et al., 2024; Chu et al., 2025), have successfuly demostrated stable learning w/ these methods. Did authors experiment with other learning rates and hyperparameter choices for different methods and observed the instability of GRPO, RLOO and other methods consistently?
- The implementation of CPGD in equation 2 (section 4.1) w/ sequence level ratios, which also formed the basis of theoretical analysis in proposition 1 and theorem 1 differs from actual implementation of their method in equation 3 (section 4.3)

**Questions:**

- Question about hyperparameter choices for the baselines (see weaknesses part of the review).

---

### Official Review · Reviewer_JxX5 · 2025-11-02

**Soundness:** 2
**Presentation:** 2
**Contribution:** 2
**Rating:** 2
**Confidence:** 3

**Summary:**

The paper focuses on tackling training instability commonly associated with RL algorithms for LMs. It identifies the instability in training and attributes it to the use of importance sampling ratios in the loss function. The authors propose CPGD, which replaces PPO loss with REINFORCE loss, uses PPO clip mechanism, a policy drift regularizer, and weighted advantages for each sample to achieve better average performance.

**Strengths:**

1. REINFORCE + clip(log-ratio) + drift KL with capped k3 gradient is easy to implement with existing RLHF codebases.

2. Demonstrated the problem of training instability among a wide range of baselines. Training stability is compared between REINFORCE++, RLOO, GRPO with or without dual clip, PG, CPG, PGD, and CPGD.

**Weaknesses:**

1. The RL community has long studied trust-region/proximal updates (TRPO/PPO), KL regularization variants, and design choices affecting PPO stability. While the paper cites PPO and related analyses, it should more explicitly compare to established stability methods and explain why REINFORCE + log-clip + KL-drift is strictly better than well-tuned PPO. Specifically, here are some problems that come to my mind:

* Conflict between clipping and KL penalty: One example is when A>0 and r > 1 + $\epsilon$. There will be no gradient from clipping, so zero policy gradient from the surrogate. Meanwhile, KL penalty pushes the policy back to $\pi_{old}$, undoing the improvement update.

* Policy update being too conservative: The clip truncates gradients on some samples, while the KL term enforces a global proximity to the old policy on all samples. Some samples will be affected by both objectives and yielding a much smaller step than intended.

* Introduce more bias into the objective: Both Clipping and KL penalty bias the policy gradient but reduce variance. Using both tricks will end up with a more biased objective compared to existing methods that use only one of the variance-reducing tricks.

* Hyperparameter tuning will be more brittle: Concurrently tuning the clipping parameter $\epsilon$ and KL weight $\alpha$ introduces extra uncertainty to the system.

2. The models are only evaluated on math benchmarks. Evaluating it on different tasks would significantly improve the soundness of the proposed method.

**Questions:**

Please see weaknesses.

Additional questions:

1. How does CPGD’s performance compare to both PPO-clip and PPO-KL penalty? PPO is an important baseline that seems to be missing from experiments.

2. Related work should include a section that focuses on existing methods that focus on training instability in RL.

---

### Note · Authors · 2026-01-06

I have read and agree with the venue's withdrawal policy on behalf of myself and my co-authors.